# Improved Swarm Intelligent Blind Source Separation Based on Signal Cross-Correlation

**DOI:** 10.3390/s22010118

**Published:** 2021-12-24

**Authors:** Jiali Zi, Danju Lv, Jiang Liu, Xin Huang, Wang Yao, Mingyuan Gao, Rui Xi, Yan Zhang

**Affiliations:** 1College of Big Data and Intelligent Engineering, Southwest Forestry University, Kunming 650224, China; zijiali@swfu.edu.cn (J.Z.); jungleliu@swfu.edu.cn (J.L.); huangxin615@swfu.edu.cn (X.H.); yaowang@swfu.edu.cn (W.Y.); TuAYuan@swfu.edu.cn (M.G.); xirui@swfu.edu.cn (R.X.); 2School of Mathematics and Physics, Southwest Forestry University, Kunming 650224, China; zhangyan@swfu.edu.cn

**Keywords:** speech separation, cross-correlation, blind source separation, swarm intelligence optimization algorithms

## Abstract

In recent years, separating effective target signals from mixed signals has become a hot and challenging topic in signal research. The SI-BSS (Blind source separation (BSS) based on swarm intelligence (SI) algorithm) has become an effective method for the linear mixture BSS. However, the SI-BSS has the problem of incomplete separation, as not all the signal sources can be separated. An improved algorithm for BSS with SI based on signal cross-correlation (SI-XBSS) is proposed in this paper. Our method created a candidate separation pool that contains more separated signals than the traditional SI-BSS does; it identified the final separated signals by the value of the minimum cross-correlation in the pool. Compared with the traditional SI-BSS, the SI-XBSS was applied in six SI algorithms (Particle Swarm Optimization (PSO), Genetic Algorithm (GA), Differential Evolution (DE), Sine Cosine Algorithm (SCA), Butterfly Optimization Algorithm (BOA), and Crow Search Algorithm (CSA)). The results showed that the SI-XBSS could effectively achieve a higher separation success rate, which was over 35% higher than traditional SI-BSS on average. Moreover, SI-SDR increased by 14.72 on average.

## 1. Introduction

Blind Source Separation (BSS) refers to the process of obtaining source signals from mixed signals when the theoretical model and source signals cannot be accurately known [1,2,3]. It is a powerful signal processing method, which has been widely used in sonar and radar signal processing [4,5], wireless communication [6], geophysical exploration [7,8], biomedical signal processing [9,10], speech and image processing [11,12], and machine fault diagnosis [13,14].

In recent years, some achievements have been made in the theoretical BSS research. BSS algorithms are mainly divided into traditional algorithms and intelligent algorithms. Independent Component Analysis (ICA) is the traditional method of linear BSS to obtain the best estimate of the source signals [15]. Since the traditional optimization algorithms are easily trapped into local optimum and have a slow convergence rate, researchers have introduced swarm intelligence optimization algorithms to ICA for blind source separation [3]. Swarm intelligence (SI) algorithm is a new branch of artificial intelligence with low cost, fast speed, and good robustness [16]. Compared with the traditional ICA methods, the BSS algorithms with the swarm intelligence optimization algorithm are characterized by higher accuracy, efficiency, and robustness [17].

The BSS based on the swarm intelligence optimization algorithm (SI-BSS) can effectively solve traditional algorithms’ nonlinear activation function selection, especially in speech separation [18]. It reduces the number of iterations of the algorithm and has global solid optimization ability. The speed of separation and convergence precision were studied in terms of their feasibility. Zhu-cheng Li et al. proposed a modified glowworm swarm optimization (MGSO) algorithm based on a novel step adjustment rule and then applied MGSO to BSS [19], which had a faster convergence speed and higher accuracy compared with PSO and GA. Fang-chen Feng et al. proposed a new optimization framework to improve the signal inference ratio without reducing the Signal to Distortion Ratio (SDR) [20]. Khalfa Ali et al. proposed the High Exploration Particle Swarm Optimization (HEPSO) algorithm for speech blind source separation with better global optimization performance [21]. Jiong Li et al. formulated a new contrast function by a convex combination of generalized autocorrelations and the statistics of the innovation, which was better than the compared algorithm in convergence speed and convergence accuracy [22]. Cristina P. Dadula and Elmer P. Dadios et al. proposed a genetic algorithm for blind source separation based on independent component analysis (ICA), effectively separating target signals. However, the experimental results also pointed out that the SI method was ineffective in completely separating the whole signals in a multi-blind source [23].

Tracing back the causes of incomplete separation, we found that each signal decomposition is independent, with no information exchange within separated signals, so each separation result does not discriminate. That is, the current separation result cannot be verified if the previous separation has separated it. Therefore, we proposed an improved blind source separation of SI algorithm based on signal cross-correlation to solve the incomplete separation. This paper mainly studies the blind source separation of linearly mixed signals with two source signals. First of all, our method tries to make more sets of separated signals with the subtraction of separated signals. Then, the cross-correlation values of all separated signals are calculated. In the end, the final separated signals are identified by the cross-correlations values, which means that the lower the values of the cross-correlations of the separated signals are, the more mutual independence they have. In order to verify the effectiveness of our algorithm, six swarm intelligence algorithms (PSO [3,24], GA [25,26], DE [27,28,29], SCA [30,31,32], BOA [33,34], and CSA [35,36]) are applied. The results of separation were compared in terms of separation performance, separation success rate, and applicability according to the cross-correlation and three kinds of separation evaluation indexes (scale invariant signal-to-distortion ratio (SI-SDR), perceptual evaluation of speech quality (PESQ), and short-time objective intelligibility (STOI) [37,38,39,40]).

## 2. Related Work

### 2.1. Blind Source Separation

The widely used BSS modeling is shown in Figure 1.

In the blind source separation, the mixed-signal X(t) waveform and the source signal’s independence are used to make the separated signal Y(t) as close to the source signal as possible. The source signal is expressed as $S\left(t\right)={\left\{s_{1}\left(t\right){,\;s}_{2}\left(t\right){,\;s}_{3}\left(t\right){,\ldots \;,\;s}_{n}\left(t\right)\right\}}^{T}$. The mixed signal is $X\left(t\right)={\left\{x_{1}\left(t\right){,\;x}_{2}\left(t\right){,\;x}_{3}\left(t\right){,\ldots \;,\;x}_{n}\left(t\right)\right\}}^{T}$. The separated signal is $Y\left(t\right)={\left\{y_{1}\left(t\right){,\;y}_{2}\left(t\right){,\;y}_{2}\left(t\right){,\ldots \ldots ,\;y}_{n}\left(t\right)\right\}}^{T}$. The mathematical model of blind source separation [1,2,3,5,16,17,19,20,21,23,41,42,43,44,45,46] is a linear mixture model, as shown in Equation (1):(1)X(t)=A×S(t),
where *A* is the linear mixing matrix of n×n, and *n* represents the number of source signals. According to the de-correlation and non-Gaussian criteria, the separation matrix *W* is solved, and then the separated signal is extracted from the X(t), which can be expressed as Equation (2):(2)Y(t)=W×X(t),
where W is the separation matrix of n×n.

### 2.2. Swarm Intelligent Blind Source Separation Algorithm

The basic idea of BSS is to separate the source signals from the mixed signals by estimating the separation matrix W. SI-BSS uses a SI algorithm to find the optimal separation matrix $W_{best}$, as shown in Figure 2.

The steps of blind source separation based on swarm intelligence algorithm are described as Algorithm 1.
**Algorithm 1**: SI-BSS (Blind Source Separation Based on Swarm Intelligence Algorithm)**Input:**X(t)Centralize and whitening X(t)Initialize the parameters of swarm intelligence algorithm.Randomly generate the initial n×n separation matrix W.Apply the SI algorithm to get $W_{best}$ according to the fitness functionObtain the separated signals Y(t), $\;Y\left(t\right)=W_{best}X\left(t\right)$**Output:**$W_{best}$, Y(t)

In order to analyze the performance and separation effect of the algorithm, simulation experiments are generally adopted. Therefore, we randomly generated the mixing matrix of n×n and mixed it with the source signals S(t) to obtain the mixed signals X(t).

Whitening: To achieve the decorrelation of mixed signals and get the unit variance, whitening is used for mixed signals X(t) as in Equation (3):(3)$$\tilde{X\left(t\right)}=ED^{-\frac{1}{2}}E^{T}X\left(t\right),$$
where E represents the eigenvector of $E\left\{XX^{T}\right\}$, and D denotes the eigenvalues of $E\left\{XX^{T}\right\}$. The aim of the whitening process is the orthogonal mixing matrix. The orthogonal mixing matrix halves the number of estimated parameters, so it only has n(n−1)/2 free parameters. By data whitening, the algorithm’s computational complexity is reduced, which leads to a high probability of achieving a successful signal recovery.

### 2.3. The Fitness Function of SI-XBSS

According to the Mean Value Theorem, the negentropy of separated signals is defined to construct the fitness function of the SI algorithm. As the negentropy is maximum, the non-Gaussian property between separated signals is also the strongest. Therefore, negentropy can be used to measure the coherence of separated signals [14]. The following formula can approximately describe multivariable negentropy.
(4)$$\;J_{i}\left(y_{i}\right)≌\;\frac{1}{12}k_{3}^{2}\left(y_{i}\right)+\frac{1}{48}k_{4}^{2}\left(y_{i}\right)+\frac{7}{48}k_{3}^{4}\left(y_{i}\right)-\frac{1}{8}k_{3}^{2}\left(y_{i}\right)k_{4}\left(y_{i}\right),$$
where $k_{3}\left(y_{i}\right)$ is the third-order cumulant of the signal, $k_{4}\left(y_{i}\right)$ is the fourth-order cumulant of the signal. When the probability distribution of the signal is symmetric, $k_{3}=0$, the above equation can be simplified to Equation (5).
(5)$$\;J_{i}\left(y_{i}\right)≌\;\frac{1}{48}k_{4}^{2}\left(y_{i}\right).$$

The kurtosis of the signal is defined as shown in Equation (6).
(6)$$k_{4}\left(y_{i}\right)=\frac{E\left(y_{i}^{4}\right)}{E{\left(y_{i}^{2}\right)}^{2}}-3.$$

## 3. Proposed Method

### 3.1. The Analysis of the Incomplete Separation

When SI algorithms were used to find the optimal $W_{best}$ to obtain the separated signals Y(t), the separation may be incomplete. To search for the reason, this paper visualized the fitness values of the two source signals, as shown in Figure 3, and the separated signals are described in Equation (7).
(7)$$\left[\begin{array}{c} y_{1}\left(t\right)\\ y_{2}\left(t\right) \end{array}\right]=Y\left(t\right)=WX\left(t\right)=\left[\begin{array}{cc} w_{1} & w_{2}\\ w_{3} & w_{4} \end{array}\right]\left[\begin{array}{c} x_{1}\left(t\right)\\ x_{2}\left(t\right) \end{array}\right],$$
where *w*_1_–*w*_4_ was in [−1,1].

In Figure 3, it is obvious that $y_{1}\left(t\right)$ and $y_{2}\left(t\right)$ have the similar fitness map, which is consistent with the principle that the separation of signal sources is independent. That means both $y_{1}\left(t\right)$ and $y_{2}\left(t\right)$ have the ability to find two signal sources. Although the fitness values of the two separated signals both reached a better area at certain position *W* searched by the SI algorithm, the two separated signals came from the same signal source.

The independent searching for the separation matrix W led to obtain the same signal source, especially when the fitness of one source is obviously higher than that of the other. Therefore, we first attempted to use the symmetric matrix to enhance the association between signals in the separation process to obtain different separated signals. The separation effect was improved slightly, as shown in Figure 4. Although symmetric *W* can reach the second separated signal source’s optimization at the constrain of *w*_1_ and *w*_2_ for the first separated signal source, the fitness map showed that the second source had significantly lower fitness, which caused the poor separation effect. This may be the defect of the fitness function itself. Therefore, enhancing the difference of the separated signals could be a way to solve the incomplete separation problem.

### 3.2. Cross-Correlation

Cross-correlation was introduced into our method. In signal processing, cross-correlation is a measure of similarity of two signals as a function of the displacement of one relative to the other [41,42].

For discrete real signal f and g, the cross-correlation is defined as:(8)$$\begin{array}{c} \hat{R_{fg}}\left(m\right)=\sum_{t=0}^{N}f\left[t\right]g\left[t+m\right],\\ N=length\left(f\left[t\right]\right)+length\left(g\left[t\right]\right)-1. \end{array}$$

The higher value of the cross-correlation of two signals represents that the more similar they are, the more dependent they are. On the contrary, the lower their cross-correlation value is, the more different and independent they are.

### 3.3. SI-XBSS Algorithm

The SI-XBSS algorithm shown in Algorithm 2 and Figure 5 mainly focuses on linearly mixing two sources of blind source separation based on the traditional SI-BSS. The SI-XBSS proposed a candidate separation pool and selected the best separated signals in the pool by cross-correlation. The candidate separation pool $Y^{\mathrm{pool}}\left(t\right)$ is composed of ${\left\{Y^{W}\left(t\right),Y^{Minus}\left(t\right),Y^{exMinus}\left(t\right)\right\}}^{T}$. $Y^{W}\left(t\right)$ is obtained from the $W_{best}$; $Y^{Minus}\left(t\right)\;$ is computed by *X*(*t*)-$Y^{W}\left(t\right)$. Due to the cocktail party effect, it is very difficult to determine the sequence of the source of the separated signals. So, we exchanged the sequence of $Y^{W}\left(t\right)$ and subtracted $Y^{W}\left(t\right)$ from *X*(*t*) to get $Y^{exMinus}\left(t\right)$. In the end, the cross-correlation values of all candidate-separated signals are calculated. The final separated signals are identified by the value of the minimum cross-correlation in the pool.
**Algorithm 2**: SI-XBSS (Swarm Intelligent Blind Source Separation based on Cross-Correlation)**Input:**X(t)Centralize and whitening X(t)Initialize the parameters of swarm intelligence algorithm.Randomly generate the initial 2×2 separation matrix W. According to the fitness function, find separation matrix $W_{best}$ by SI algorithms.Obtain mixed signals $Y^{W}\left(t\right)$$Y^{W}\left(t\right)={\left\{\;y_{1}^{w}\left(t\right),y_{2}^{w}\left(t\right)\right\}}^{T}=W_{best}X\left(t\right)$Obtain mixed signals $Y^{Minus}\left(t\right)\;$$Y^{Minus}\left(t\right)=\left[\begin{array}{c} x_{1}\left(t\right)\\ x_{2}\left(t\right) \end{array}\right]-\left[\begin{array}{c} y_{1}^{w}\left(t\right)\\ y_{2}^{w}\left(t\right) \end{array}\right]=\left[\begin{array}{c} y_{1}^{Minus}\left(t\right)\\ y_{2}^{Minus}\left(t\right) \end{array}\right]$Obtain mixed signals $Y^{exMinus}\left(t\right)$$Y^{exMinus}\left(t\right)=\left[\begin{array}{c} x_{1}\left(t\right)\\ x_{2}\left(t\right) \end{array}\right]-\left[\begin{array}{c} y_{2}^{w}\left(t\right)\\ y_{1}^{w}\left(t\right) \end{array}\right]=\left[\begin{array}{c} y_{1}^{exMinus}\left(t\right)\\ y_{2}^{exMinus}\left(t\right) \end{array}\right]$Obtain the candidate separation pool $Y^{\mathrm{pool}}\left(t\right)$$\begin{array}{cc} Y^{\mathrm{pool}}(t) & ={\left\{y_{1}^{pool}(t),y_{2}^{pool}(t),y_{3}^{pool}(t),y_{4}^{pool}(t),y_{5}^{pool}(t),y_{6}^{pool}(t)\right\}}^{T}\\ & ={\left\{Y^{W}(t),Y^{Minus}(t),Y^{exMinus}(t)\right\}}^{T}\\ & ={\left\{y_{1}^{W}(t),y_{2}^{W}(t),y_{1}^{Minus}(t),y_{2}^{Minus}(t),y_{1}^{exMinus}(t),y_{2}^{exMinus}(t)\right\}}^{T} \end{array}$Calculate Xcorr ($y_{i}^{pool}(t),y_{j}^{pool}(t))$$i,j\in \left[1,\lambda \right],i\neq j,\;\lambda =total\;number\;of\;Y^{\mathrm{pool}}(t)$$Y(t)=\min\;(\mathrm{Xcorr}\;(y_{i}^{pool}(t),y_{j}^{pool}(t)))$**Output:**$W_{best}$, Y(t)

In the SI-XBSS algorithm, we mainly studied mixed signals of two source signals. Therefore, the mixed signals were obtained in the simulation experiment by mixing two known source signals with a randomly generated 2 × 2 mixture matrix.

### 3.4. Success Rate of Separation

In order to evaluate the effectiveness of the separated signals and compare the separation performance of the SI-XBSS, the success rate of separation was defined as shown in Equation (9).
(9)$$\;SuccessRate_{g}=\mathrm{Mean}\left(\sum_{g=1}^{G}\frac{\;\sum_{r=1}^{R}Number_{r}}{round_{g}}\right),$$
where G is the number of total mixed signals, and *R* is the total executed round for each mixed signal. $Number_{r}$ is the round of complete separations at each mixed signal. $round_{g}$ is a total round of the *g*^th^ group. $SuccessRate_{g}$ is the rate of complete separation at each mixed signal.

## 4. Experiments

### 4.1. Data and Platforms

The simulation experiments were designed by MATLAB 2019B. The experimental data (Appendix A) are 50 speech segments of females, and 50 speech segments of males, both in the MOCHA-TIMIT corpus database. Each segment of signal lasted 1 s with 16 kHz as the sampling frequency.

### 4.2. Experimental Setup

Two segments generated a test group of mixture signals, where one group was female and the other group was male, which were linearly mixed with random *A* matrix. The experiment set 50 mixed test groups. Our IS-XBSS was applied in six kinds of swarm intelligent algorithms (PSO, GA, DE, SCA, BOA, and CSA) and compared with IS-BSS in these six algorithms. The separation quality was evaluated with SI-SDR, STOI, and PESQ.

The parameters of six swarm intelligent algorithms were set as NP = 200, iteration = 50. Each group of mixed signal is executed for 50 runs in each algorithm. The test mixture signal groups were 50.

### 4.3. Results

The following four parts presented the results: (1) overall effect of separation, (2) results of complete separation, (3) results of incomplete separation, and (4) results of non-separation.

#### 4.3.1. Overall Separation Performance

The overall results are shown in Table 1 and Figure 6. A total of 31 groups of mixed signals can be completely separated by the SI-XBSS, including 14 groups from the incomplete and 10 groups from the non-separation by SI-BSS. The incomplete separation groups were decreased from 20 groups by SI-BSS to 17 groups by our SI-XBSS. In addition, the non-separation groups decreased significantly from 23 groups by SI-BSS to 2 groups by SI-XBSS. The results presented that our SI-XBSS significantly increased the groups of successful separation, effectively reduced the number of non-separated groups, and significantly improved the separation performance.

The success rate of separation, defined as Equation (9), of the SI-XBSS with six optimization algorithms, is shown in Figure 7. The separation success rate of SI-XBSS exceeded 35% of that of SI-BSS, among which the PSO-XBSS improved 43.32%, which was the most significant improvement.

The mean evaluation values of the six SI-BSS and the six SI-XBSS algorithms are shown in Table 2. Compared with SI-BSS, SI-XBSS increased by 13.83 to 17.83 for SI-SDR, 0.17 to 0.19 for STOI, and 0.01 to 0.03 for PESQ. The results show that the proposed method has significantly improved the evaluation values of the three evaluation indexes.

The separated waves of signals are shown in Figure 8. Obviously, the traditional PSO-BSS only separated one of the two mixed signals in Figure 8c, while our PSO-XBSS separated two signals in Figure 8d.

The mean values of cross-correlations of the source signals and the separated signals by SI-BSS and SI-XBSS are shown in Figure 9. The results showed that the source signals had the lowest cross correlation value—less than 20; the value by SI-XBSS was very close to that of the source signals with a range from 55.20 to 58.70; and the separated signals by the SI-BSS was from 180.98 to 182.00, which was more than four times of that of the source signals. It could be inferred that the smaller the cross-correlation value between signals was, the lower the similarity of the signals was. For the separated signals, the cross-correlation value can reflect the difference of separated signals, which means the lower the value of cross-correlation was, the better the separation performance was.

The convergence speed of SI-XBSS is shown in Figure 10. The results showed that all the six SI-XBSS were converged. The GA-XBSS had the slowest convergence speed among the six algorithms but with the highest fitness. PSO-XBSS and CSA-XBSS had the fastest convergence speed with the lowest fitness.

SI-SDR, PESQ, and STOI were taken to evaluate the separation performance. The mean value of the three evaluation indexes is shown in Table 3. The SI-SDR increased by 50.33–59.85, and STOI increased from about 0.23 to above 0.99. It showed that the improved method effectively enhanced the evaluation values of SI-SDR and STOI.

#### 4.3.2. The Results of Complete Separation

The stability of the complete separated signals by SI-XBSS and SI-BSS is shown in Figure 11. A total of 31 groups in 50 groups achieved complete separation with our method. In 50 rounds of separation, from the 1st to the 24th group, the complete separation rounds of the SI-BSS of PSO, GA, DE, SCA, BOA, and CSA were all 0. However, the complete separation rounds of the SI-XBSS of PSO, GA, DE, SCA, BOA, and CSA ranged from 15 to 50 rounds. From the 25th to the 31st group, all improved algorithms achieved complete separation in all 50 rounds. The results showed that our algorithm significantly increased the rounds of complete separation.

The mean cross-correlation values of separated signals of PSO-BSS, BOA-BSS, PSO-XBSS, and BOA-XBSS from the 1st to the 31st group are shown in Figure 12. The results showed that the cross-correlation values of the PSO-XBSS and BOA-XBSS were lower than those of PSO and BOA from 1st to the 31st group and were closer to that of the source signals. The results of other SI-BSS and SI-XBSS (GA, DE, SCA, and CSA) were consistent with BOA. We found that when the value of cross-correlation of the separated signals was close to that of the source signals, it had a high probability of being completely separated.

#### 4.3.3. The Result of Incomplete Separation

Table 4 showed the improvement of the mean value of SI-SDR for $y_{1}$ and $y_{2}$ of the incomplete 17 groups. The SI-SDR value of $y_{1}$ was increased in groups 2, 3, 5, 6, 8, 9, and 10. For $y_{2}$, the SI-SDR value was increased in groups 1, 4, 7, and 11–17 and decreased slightly in group 10. The results showed that the SI-XBSS improved the evaluation value of SI-SDR to enhance the quality of the separated signal in the incomplete separation.

#### 4.3.4. Unseparated Results

Two groups in the 50 mixed groups were not yet separated by our method; their SI-BSS are shown in Table 5. The results showed that although the two groups were not separated successfully, the separation performance was still significantly improved from −13.3 to 10.1 for $y_{1}$ in one mix-group and from −2.6 to 6 around for $y_{2}$ in another mix-group.

## 5. Discussion

According to the results of previous studies [1,3,23,45,46,47,48,49,50,51,52,53,54,55,56], the quality of speech blind source separation could be improved by swarm intelligence algorithms, but it is rare to find a study about multi-groups with random linear mixed signals. In their studies, the quality, convergence speed, and convergence accuracy of BSS were significantly improved by enhancing the swarm intelligence optimization algorithm. However, their methods only analyzed the effectiveness and separability of one or five mixed signals groups with their algorithms. In order to explore the separation effectiveness of multi- groups of mixed signals based on SI, our paper implements six intelligent optimization algorithms (PSO, GA, DE, SCA, BOA, and CSA) and carries out a large number of experiments on speech blind source separation. We found that incomplete separation often occurs in SI-BSS through the comprehensive analysis of experimental results. In order to solve this problem, the signal cross-correlation was introduced to our study to form SI-XBSS. Experimental results showed that the SI-XBSS has higher effectiveness of speech blind source separation in a large number of mixed signals than SI-BSS has.

SI-XBSS has three advantages from the above results: first, it has effectively improved the separation success rate compared with the SI-BSS algorithm. Second, the proposed algorithm can still improve the performance and quality of separated signals when they cannot be separated entirely. Third, it has strong stability and applicability. We applied six kinds of SI (PSO, GA, DE, SCA, BOA, and CSA) for speech blind source separation in 50 test mixed signals with 50 runs for each SI. It showed that the success rates were improved above 35% on average, and SI-SDR increased by 14.72 with the SI-XBSS.

Although SI-XBSS has three advantages, it can still improve the separation success rate and performance. The SI-XBSS has not fully solved the problem of the incomplete separation, as we still can find some incomplete separated signals. Furthermore, although the SI-XBSS is mainly aimed at linear mixed double-source speech blind source separation, its improvement effect is relatively significant, and it also provides a direction for the subsequent improvement of multi-source speech blind source separation. We also tried to add cross-correlation into the fitness function to improve the fitness function. However, the experimental results are consistent with those of the SI-XBSS, but it took more time to calculate and SI-XBSS did. In order to obtain a better separation performance and higher success rate, our future work will improve the fitness function combined with cross-correlation.

## 6. Conclusions

In this paper, we proposed a blind source separation of the swarm intelligence optimization algorithm based on the cross-correlation of separated signals (SI-XBSS) to solve the incomplete separation. A large number of mixed signals were executed by SI-XBSS. Experimental results showed that the SI-XBSS could effectively improve the success rate of the separation and increase the SI-SDR, PESQ, and STOI at a different level. Furthermore, it has strong stability and applicability in different SI.

## Figures and Tables

**Figure 1 sensors-22-00118-f001:**
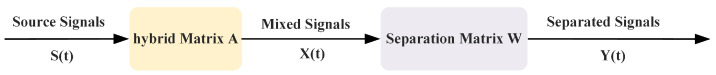
BSS modeling flow chart.

**Figure 2 sensors-22-00118-f002:**
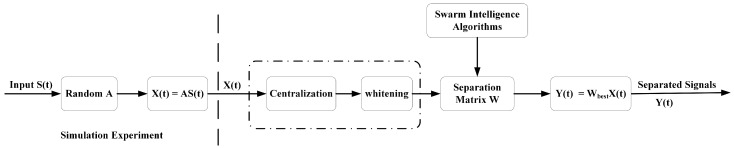
SI-BSS flow chart.

**Figure 3 sensors-22-00118-f003:**
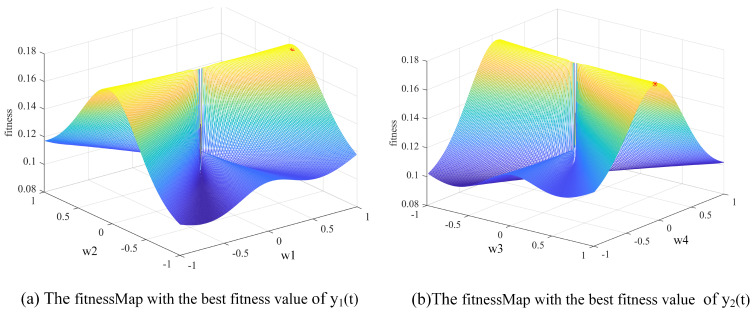
The optimum fitness range of $y_{1}\left(t\right)$ and $y_{2}\left(t\right)$. The red dots represent the best fitness values at w position, which was searched by the swarm intelligence algorithms. When [$w_{1}$, $w_{2}$] is [0.93, −0.13] searched by SI, $y_{1}\left(t\right)$ can reach the fitness of 0.17; When [$w_{3}$, $w_{4}$] is [1, −0.06] searched by SI, $y_{2}\left(t\right)$ can reached the fitness of 0.18.

**Figure 4 sensors-22-00118-f004:**
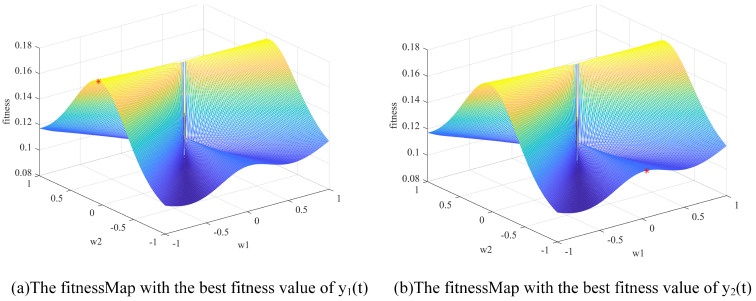
The optimum fitness range of $y_{1}$ and $y_{2}$ using the symmetric matrix. The red dots represent the best fitness values at w position, which was searched by the swarm intelligence algorithms. When [$w_{1}$, $w_{2}$] is [−1, 0.05] searched by SI, $y_{1}\left(t\right)$ reached the fitness of 0.18; When [$w_{3}$, $w_{4}$] is [0.05, −1] searched by SI, $y_{2}\left(t\right)$ reached the fitness of 0.18.

**Figure 5 sensors-22-00118-f005:**
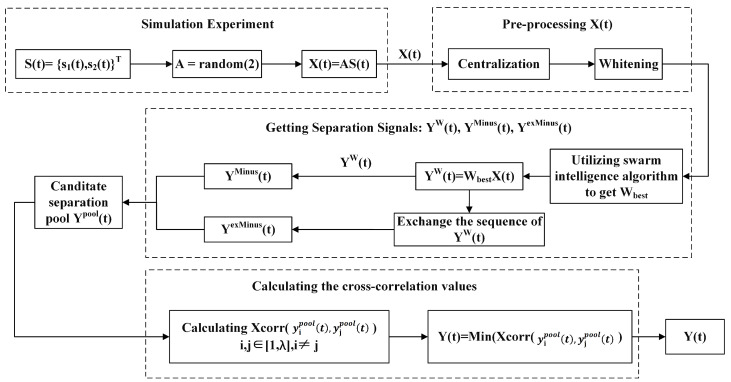
SI-XBSS flow chart.

**Figure 6 sensors-22-00118-f006:**
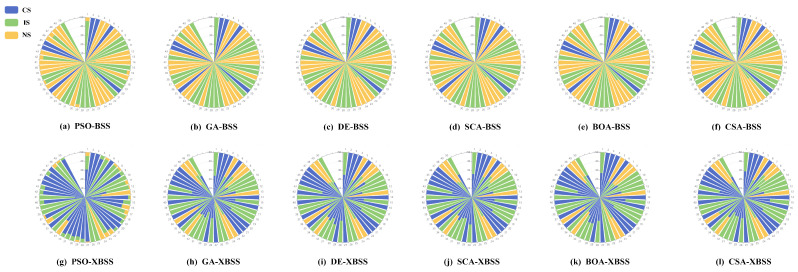
Separation results of 50 groups of mixed signals: (**a**–**f**) the SI-BSS based on PSO, GA, DE, SCA, BOA, and CSA. (**g**–**l**) the SI-XBSS of PSO, GA, DE, SCA, BOA, and CSA.

**Figure 7 sensors-22-00118-f007:**
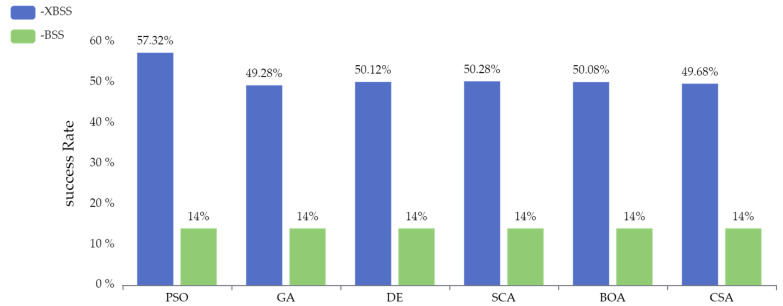
The success rate of separation of the six SI-XBSS and six SI-BSS algorithms.

**Figure 8 sensors-22-00118-f008:**
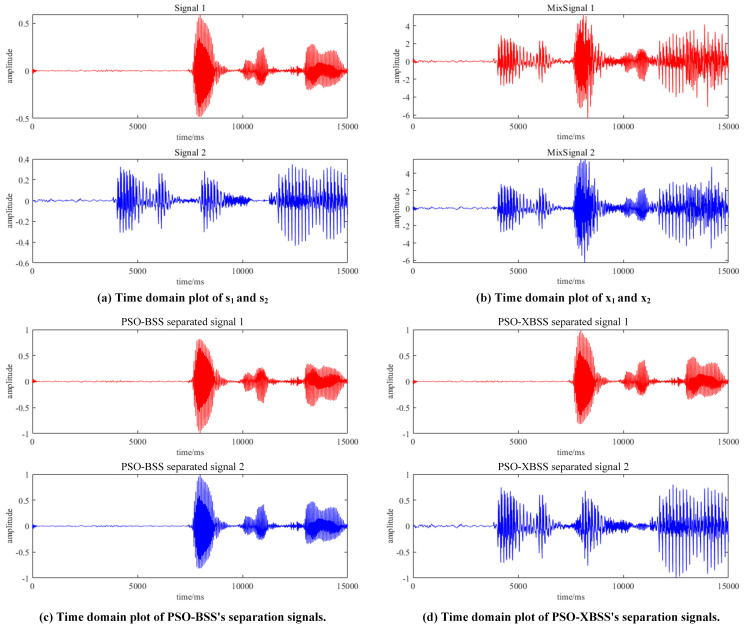
The time domain diagrams of source signals, mixed signals, and separated signals of PSO-BSS and PSO-XBSS. ($s_{1}$ represents the first source signal, and $s_{2}$ represents the other source signal. $x_{1}$ represents the first mixed signal, and $x_{2}$ represents the other mixed signal.).

**Figure 9 sensors-22-00118-f009:**
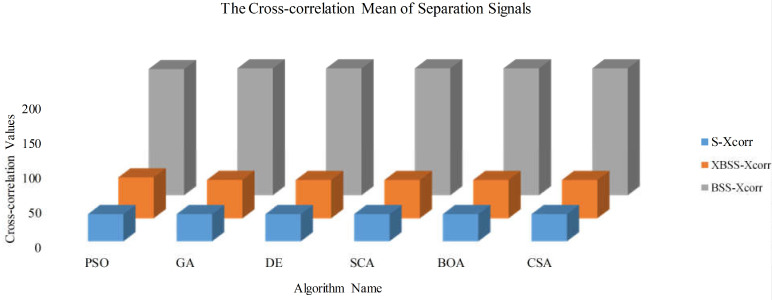
The mean cross-correlation value of separated signals obtained by the SI-BSS and SI-XBSS. (S-Xcorr was the cross-correlation value of source signals. XBSS-Xcorr was the cross-correlation value of our method, and BSS-Xcorr was the cross-correlation value of the SI-BSS.).

**Figure 10 sensors-22-00118-f010:**
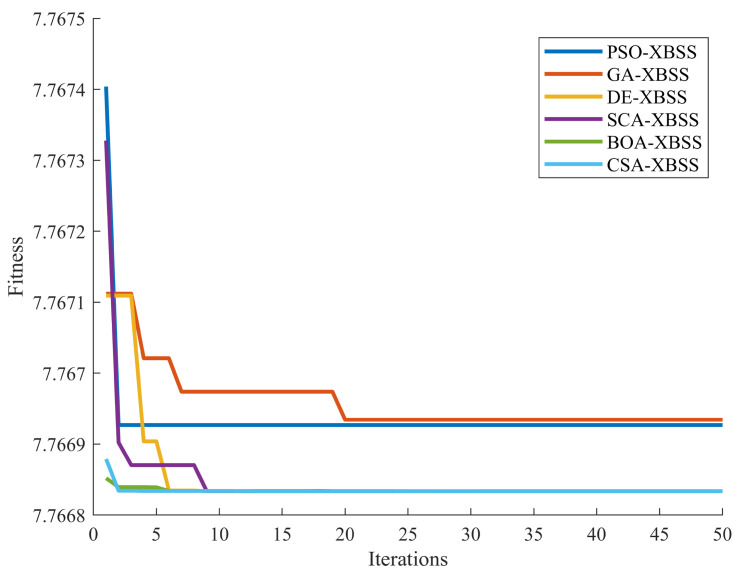
Convergence speed of the six algorithms.

**Figure 11 sensors-22-00118-f011:**
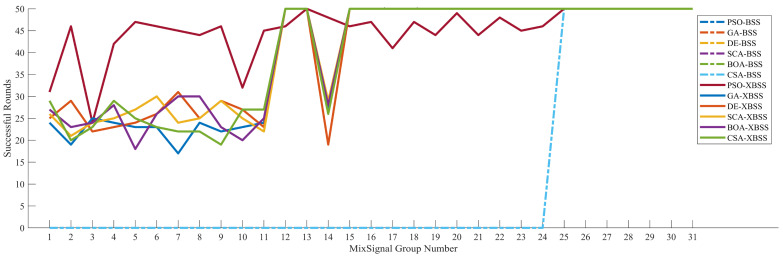
The stability of the complete separated of 12 algorithms. The results of the SI-BSS of PSO, GA, DE, SCA, BOA, and CSA are the same, so the figure shows the same dot line.

**Figure 12 sensors-22-00118-f012:**
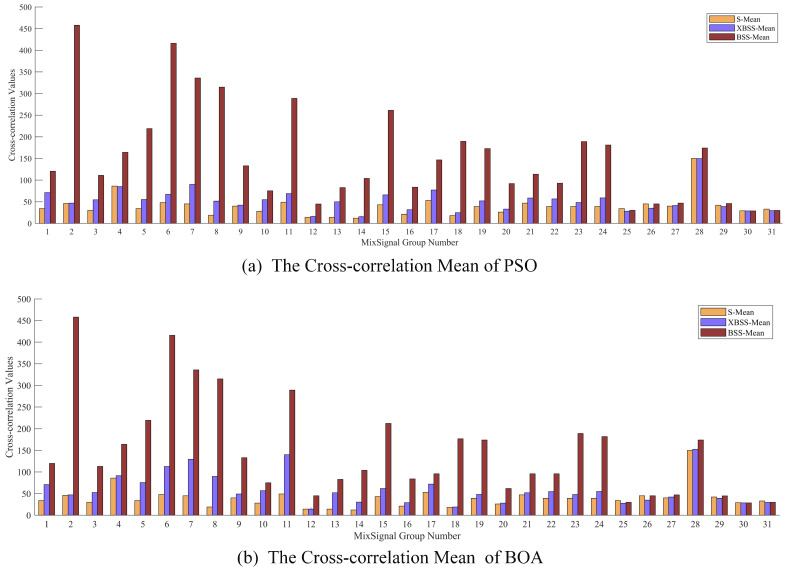
The mean cross-correlation between PSO and BOA. S-Mean is the cross-correlation between source signals. XBSS-Mean is the mean cross-correlation of separated signals obtained by PSO-XBSS and BOA-XBSS. BSS-Mean is the mean of cross-correlation between separated signals obtained by BSS based on PSO-BSS and BOA-BSS.

**Table 1 sensors-22-00118-t001:** Separation results of 50 groups mixed signals. (CS is complete separation. IS is incomplete separation. NS is non-separation. In column Complete Separation, the “14→IS” represents that 14 groups were completely separated by SI-XBSS, which were incompletely separated by SI-BSS.).

Algorithm Name	Complete Separation	Incomplete Separation	Non-Separation
PSO-BSS	7	20	23
GA-BSS	7	20	23
DE-BSS	7	20	23
SCA-BSS	7	20	23
BOA-BSS	7	20	23
CSA-BSS	7	20	23
PSO-XBSS	31 (7→CS,14→IS, 10→NS)	17 (6→IS, 11→NS)	2 (2→NS)
GA-XBSS	31 (7→CS,14→IS, 10→NS)	17 (6→IS, 11→NS)	2 (2→NS)
DE-XBSS	31 (7→CS,14→IS, 10→NS)	17 (6→IS, 11→NS)	2 (2→NS)
SCA-XBSS	31 (7→CS,14→IS, 10→NS)	17 (6→IS, 11→NS)	2 (2→NS)
BOA-XBSS	31 (7→CS,14→IS, 10→NS)	17 (6→IS, 11→NS)	2 (2→NS)
CSA-XBSS	31 (7→CS,14→IS, 10→NS)	17 (6→IS, 11→NS)	2 (2→NS)

**Table 2 sensors-22-00118-t002:** The mean evaluation values of SI-SDR, PESQ, and STOI.

Algorithm Name	SI-SDR	PESQ	STOI
PSO-BSS	6.41	1.18	0.75
GA-BSS	6.05	1.18	0.74
DE-BSS	6.00	1.19	0.74
SCA-BSS	6.17	1.19	0.74
BOA-BSS	6.02	1.19	0.74
CSA-BSS	5.99	1.19	0.74
PSO-XBSS	23.80	1.20	0.94
GA-XBSS	19.89	1.20	0.90
DE-XBSS	20.25	1.20	0.91
SCA-XBSS	20.42	1.20	0.91
BOA-XBSS	20.47	1.20	0.91
CSA-XBSS	20.15	1.21	0.91

**Table 3 sensors-22-00118-t003:** Comparison of the evaluation mean values of SI-SDR, PESQ, and STOI. ($y_{1}$ represents the first separated signal, $y_{2}$ represents the other separated signal).

Algorithm Name	SI-SDR		PESQ		STOI	
	$y_{1}$	$y_{2}$	$y_{1}$	$y_{2}$	$y_{1}$	$y_{2}$
PSO-BSS	37.30	−32.38	1.24	1.18	0.99	0.29
GA-BSS	46.49	−34.39	1.26	1.14	1.00	0.23
DE-BSS	45.19	−34.40	1.26	1.14	1.00	0.23
SCA-BSS	45.19	−34.40	1.26	1.14	1.00	0.23
BOA-BSS	45.18	−34.40	1.26	1.14	1.00	0.23
CSA-BSS	45.19	−34.40	1.26	1.14	1.00	0.23
PSO-XBSS	36.98	17.94	1.24	1.10	0.99	0.97
GA-XBSS	21.72	25.47	1.10	1.35	0.96	0.99
DE-XBSS	45.19	17.4	1.26	1.10	1.00	0.96
SCA-XBSS	45.19	17.40	1.26	1.10	1.00	0.96
BOA-XBSS	45.19	17.40	1.26	1.10	1.00	0.96
CSA-XBSS	45.19	17.40	1.26	1.10	1.00	0.96

**Table 4 sensors-22-00118-t004:** Improvement details of mean value of SI-SDR of $y_{1}$ and $y_{2}$ in groups 1 to 17.

	$y_{1}$						$y_{2}$					
X No	PSO-XBSS	GA-XBSS	DE-XBSS	SCA-XBSS	BOA-XBSS	CSA-XBSS	PSO-XBSS	GA-XBSS	DE-XBSS	SCA-XBSS	BOA-XBSS	CSA-XBSS
1	0.01	0.01	0.00	0.00	0.00	0.00	**77.64**	**46.10**	**40.35**	**46.09**	**37.48**	**40.35**
2	**53.65**	**37.41**	**37.40**	**37.40**	**28.38**	**39.21**	0.00	0.02	0.00	0.00	0.00	0.00
3	**42.09**	**20.63**	**21.31**	**26.86**	**23.39**	**29.63**	0.02	0.00	0.00	0.00	0.00	0.00
4	0.06	−0.02	0.00	0.00	0.00	0.00	**45.56**	**28.10**	**30.79**	**37.35**	**37.72**	**37.35**
5	**40.83**	**45.74**	**45.81**	**45.81**	**45.80**	**45.81**	1.36	−0.19	0.00	0.00	0.00	0.00
6	**36.91**	**41.37**	**41.33**	**41.33**	**41.33**	**41.33**	1.41	0.00	0.00	0.00	0.00	0.00
7	6.37	0.00	0.00	0.00	0.00	0.00	**29.51**	**35.40**	**35.38**	**35.38**	**35.37**	**35.38**
8	**44.93**	**5.68**	**14.21**	**9.10**	**12.51**	**12.51**	0.00	0.02	0.00	0.00	0.00	0.00
9	**29.24**	**20.90**	**22.64**	**21.94**	**21.24**	**23.33**	−0.01	−0.06	0.00	0.00	0.00	0.00
10	**5.09**	**3.71**	**4.13**	**3.44**	**3.44**	**3.44**	−2.14	−1.23	−1.89	−1.56	−1.56	−1.56
11	4.87	−0.01	0.00	0.00	0.00	6.21	**40.45**	**42.56**	**42.57**	**42.57**	**42.57**	**40.85**
12	0.00	0.01	0.00	0.00	0.00	0.00	**41.50**	**24.42**	**22.86**	**29.08**	**24.42**	**33.74**
13	0.02	0.03	0.00	0.00	0.00	0.00	**39.25**	**30.72**	**32.96**	**29.24**	**31.84**	**31.84**
14	0.00	0.01	0.00	0.00	0.00	0.00	**40.68**	**16.20**	**23.99**	**17.32**	**19.54**	**17.32**
15	2.32	0.00	0.00	0.00	0.00	0.00	**35.85**	**36.42**	**36.42**	**36.42**	**36.42**	**36.42**
16	0.00	0.00	0.00	0.00	0.00	0.00	**27.11**	**17.24**	**18.33**	**18.70**	**18.33**	**17.97**
17	0.01	0.01	0.00	0.00	0.00	0.00	**28.90**	**16.81**	**10.77**	**10.77**	**16.14**	**10.77**

The data in bold are the improved SI-SDR values of separated signals obtained by SI-XBSS.

**Table 5 sensors-22-00118-t005:** The estimated value of SI-SDR of the two groups.

			SI-BSS			SI-XBSS		
X No	S-Xcorr	**Algorithm**	Y-Xcorr	$\mathrm{SI}\text{-}\mathrm{SDR}y_{1}$	$\mathrm{SI}\text{-}\mathrm{SDR}y_{2}$	Y-Xcorr	$\mathrm{SI}\text{-}\mathrm{SDR}y_{1}$	$\mathrm{SI}\text{-}\mathrm{SDR}y_{2}$
**1**	28	PSO	180.3	−13.3	13.9	**104.6**	**10.1**	13.9
		GA	165	−13.8	14.3	**117.6**	**10.1**	14.3
		DE	165	−13.8	14.3	**118**	**10.1**	14.3
		SCA	165	−13.8	14.3	**118**	**10.1**	14.3
		BOA	165	−13.8	14.3	**118**	**10.1**	14.3
		CSA	165	−13.8	14.3	**118**	**10.1**	14.3
**2**	42	PSO	478	2.8	−2.6	**176.9**	**2.9**	**7.2**
		GA	478	2.8	−2.6	**198.9**	**2.9**	**5.9**
		DE	478	2.8	−2.6	**199.9**	**2.9**	**5.9**
		SCA	478	2.8	−2.6	**199**	**2.9**	**5.9**
		BOA	478	2.8	−2.6	**203**	**2.9**	**5.6**
		CSA	478	2.8	−2.6	**195**	**2.9**	**6.1**

The data in bold represent improvements using the SI-XBSS algorithm.

## Data Availability

Not applicable.

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
