# Peer review of "Improved Swarm Intelligent Blind Source Separation Based on Signal Cross-Correlation"

_sensors, 2021, doi:10.3390/s22010118_

Round 1
Reviewer 1 Report
This paper proposes an enhancement to swarm intelligence blind source separation (SI-BSS) based algorithms with the use of post processing steps. These post processing steps are applied after obtaining the separation matrix W_{nbest} and consist in using cross-correlation of the initially separated signals Y(t) in order to further estimate them.
The idea is technically sound and the experimental results show improvements to the baseline technique without cross-correlation processing. My only concern on the paper is that it lacks references and comparison with similar approaches. If this paper gets accepted for publication I would strongly recommend the authors to add such information. Are there similar approaches out there? If so, how do they perform?
Other minor comments/suggestions are below:
- maybe one section should be specifically dedicated to explain the proposed method, and another section to be dedicated to the experiments/comparison with baselines. Section 2 mixes analyses, proposed method description and experimental set-up.
- captions of figures should include more information on what is presented. Specially in figures 3 and 4.
- English needs improvements, such as in the first sentence of Section 2.4.2.
- In Section 2.6, are there references to the metric used? Is that a standard metric?
Reviewer 2 Report
* pg.1, 18 - probably it should be "the separated signals are..." instead of "separation signals"
* pg.1, 20 - as above
* pg.1, 24 - please consider "which was over 28% higher than ..." instead "which was over 28% than"
* pg.2, 47 - "[18]. It reduced ..." should be "[18]. It reduces"
* pg.2, 54 - "AliKHALFA" why capital letters?
* pg.2, 58 - consider "than the compared algorithm" instead of "than the comparison algorithm"
* pg.3, Algorithm1 - what is the size of matrix $W$? How do you choose $Nbest$ Is matrix $W$ constant over the period of the experiment?
* pg.3, l110 - please consider changing "source separation at each time ..." to "source separation whenever ..."
* pg.3, Fig.2 - there is a lack of explanation of how do you "whitening" signals? what kind of algorithm is applied?
* pg.4, l129 - it is not clear what do you mean "the values of $t_1$ and $t_2$ of signals"? Is symbol $t$ referring time or value of the signal? What is more important in equation (3) you assumed stationary signals, thus it is enough to shift in time only one signal.
* pg.4, eq(3) - please consider using discrete cross-correlation because - as I understand - you utilize discrete signals
* pg.5, Fig.5, Algorithm.2 - do you assume that there exist only two source signals in the mixture $X(t)$? Once again what is the size of $W$? and how do you find/setup it? The sizes of matrices are not clear.
* pg.5, Fig.5, Algorithm.2 - you should explain what does the $*$ (star) symbol stands for. How it is related to $Minus$ and $ex-Minux$
* pg.5, Fig.5, Algorithm.2 - how do you guess that $\lambda$ should equal 6?
* pg.8, l200 - please, consider using standard equation references type i.e. "equation/formula (8)" instead of "Formula 8"
* pg.9, l221 - "had the lowest value less than" ==> "had the lowest cross correlation value less than"
* pg.10, Figure 10 - please, explain the symbol $e$
* pg.10, l239 - "The evaluation of separated qualities" the title is hard to catch the meaning. Does it mean the evaluation of separated signal qualities or the evaluation of separation qualities?
* pg.10, l245 - probably you mean "the ... separated signal" not "separation signal"
* pg.11, l254 - "That meant each round could completely separate the mix signals." - no idea what do you mean
* pg.11, Figure 12 - Please, place the Figure in a way that a caption wouldn't be divided by pages. It can confuse the readers.
* pg.12, l286 - "The evaluation of incomplete separated qualities" the tile is hard to understand.
General question:
1) What if the mixing of source signals is non-linear? Then, the eq. $X(t)=A*S(t)$ no longer describes a model of a process.
Nevertheless, it was sometimes hard to catch the meaning of the sentences. And even despite my language ignorance, I was able to find a number of mistakes.
The paper - IMO - presents only a small improvement when one compares the proposed algorithm to the existing one. Moreover, the proposed quality measure is not clear, so the results presented in the paper are questionable.
Round 2
Reviewer 2 Report
Good work. Now the paper is more clear and easy to understand.
The only observation is that several Figures and Tables are wider than the column width. Please check if it is allowable in the formatting style.